# TRAINING FOR COMPOSITIONAL SENSITIVITY REDUCES DENSE RETRIEVAL GENERALIZATION

**Radoslav Ralev, Aditeya Baral, Iliya Zhechev, Jen Agarwal & Srijith Rajamohan**
Redis, Bulgaria and Redis, USA
`{firstname.lastname}@redis.com`

## ABSTRACT

Dense retrieval compresses texts into single embeddings ranked by cosine similarity. While efficient for recall, this interface is brittle for identity-level matching: minimal compositional edits (negation, role swaps) flip meaning yet retain high similarity. Motivated by geometric results for unit-sphere cosine spaces (Kang et al., 2025), we test this retrieval-composition tension in text-only retrieval. Across four dual-encoder backbones, adding structure-targeted negatives consistently *reduces* zero-shot NanoBEIR retrieval (8–9% mean nDCG@10 drop on small backbones; up to 40% on medium ones), while only partially improving pooled-space separation. Treating pooled cosine as a recall interface, we then benchmark verifiers scoring token–token cosine maps. MaxSim (late interaction) excels at reranking but fails to reject structural near-misses, whereas a small Transformer over similarity maps reliably separates near-misses under end-to-end training. [1]

## 1 INTRODUCTION

The dominant dual-encoder paradigm compresses texts into fixed vectors for efficient maximum inner product search (MIPS) retrieval (Reimers & Gurevych, 2019; Karpukhin et al., 2020). While effective for fuzzy topical matching, this architecture suffers a fundamental "resolution loss" regarding composition. Because the embedding function compresses variable-length reasoning into a single point, it often treats sentences as commutative bags-of-words, struggling to distinguish *structural near-misses* (e.g.,"the dog bit the man" vs. "the man bit the dog") (Yuksekgonul et al., 2022).

Recent theory suggests this is geometrically inevitable: Kang et al. (2025) argue that unit-sphere cosine spaces force conceptual clusters into linear superposition, a geometry hostile to non-commutative structures like negation or order. This implies a *retrieval–composition tension*: forcing compositional sensitivity into a single vector degrades broad topical generalization.

**Contributions.** We investigate this tension in text-only retrieval. We show that training with structure-targeted hard negatives creates a zero-sum game: the model rejects specific permutations but suffers significant degradation in out-of-domain retrieval (NanoBEIR). We argue that identity-sensitive matching should instead be treated as a distinct *verification* task. We benchmark lightweight verifiers on token–token similarity maps, finding that while MaxSim excels at relevance, true identity preservation requires learned verifiers that detect topological patterns in the map.

## 2 SINGLE-VECTOR COSINE IS A BOTTLENECK FOR IDENTITY

Under unit-norm pooled embeddings and cosine scoring, a single inner product must simultaneously encode topical similarity and compositional distinctions. Previous work asserts that nontrivial content grouping pressures the representation toward (approximately) additive superposition (Kang et al., 2025), which is commutative and tends to erase binding/order information. This predicts brittleness: there exist minimally edited near-misses (binding swaps, role reversals, scoped negation flips) that cannot be uniformly separated from paraphrases by a fixed cosine margin under the pooled-cosine bottleneck. We include the formal assumptions and an expanded statement in Appendix B.

---

[1]Code and datasets are available at `https://github.com/radoslavralev/limitations-text-retrieval`

Table 1: Mean NanoBEIR retrieval performance (nDCG@10 and Acc@1). Model A: standard fine-tuning. Model B: + structured negatives.

| Backbone | nDCG@10 | | | Acc@1 | | |
|---|---|---|---|---|---|---|
| | Model A | Model B | Δ (% drop) | Model A | Model B | Δ (% drop) |
| MiniLM-L6 | 0.439±0.000 | 0.401±0.001 | **-0.038 (-8.7%)** | 0.393±0.002 | 0.346±0.004 | **-0.047 (-12.0%)** |
| MiniLM-L12 | 0.467±0.001 | 0.424±0.005 | **-0.043 (-9.2%)** | 0.424±0.003 | 0.369±0.010 | **-0.055 (-13.0%)** |
| GTE-Small | 0.481±0.002 | 0.442±0.006 | **-0.039 (-8.1%)** | 0.444±0.001 | 0.389±0.004 | **-0.055 (-12.4%)** |
| GTE-ModernBERT-base | 0.543±0.001 | 0.324±0.018 | **-0.219 (-40.3%)** | 0.493±0.005 | 0.275±0.015 | **-0.218 (-44.2%)** |

We adopt the standard two-stage setup. **Stage 1:** retrieve top-$K$ candidates using ANN over pooled cosine keys. **Stage 2:** verify candidates using token interactions.

Given token embeddings for query $q$ and candidate $c$, we form the token similarity map $M_{ij}(q,c) = \cos(q_i, c_j)$. A verifier $F(q,c)$ consumes $M$ (optionally with positional bias) and outputs a scalar used to rerank or gate candidates. We study a spectrum from simple reductions (global average; MaxSim/late interaction) to small learned pattern recognizers over $M$ (tiny CNN / tiny Transformer). Full definitions (including alignment-biased variants and architectures) are in Appendix C.

## 3 EXPERIMENTS

Our analysis predicts a *retrieval–composition tension* for pooled-cosine dual encoders: allocating representational margin to reject meaning-changing near-misses can reduce the margin available for coarse content grouping. We test: (i) whether structure-targeted hard negatives degrade out-of-domain retrieval, and (ii) what verifier capacity is required to reject structural near-misses. For more information on dataset generation see Appendix D.1.

### 3.1 DO COMPOSITION-SENSITIVE NEGATIVES HURT RETRIEVAL?

We fine-tune dual encoders on NQ triplets using SentenceTransformers' MultipleNegativesRankingLoss. We compare: **Model A (baseline)** trained on standard NQ supervision, and **Model B (structured)** trained on the mixed dataset described in §D.1 (standard + structural negatives). To compare across backbones under a fixed compute budget, we fix wall-clock training time per backbone and set steps based on measured throughput (details in Appendix). We evaluate zero-shot retrieval on NanoBEIR using nDCG@10 and Acc@1 (mean across datasets). Table 1 summarizes mean results across four backbones.

**Results.** Across all backbones and metrics, training with structural hard negatives (**Model B**) reduces NanoBEIR performance relative to the NQ-only baseline (**Model A**). On MiniLM-L6/L12 and `gte-small`, mean nDCG@10 drops by 8–9% and Acc@1 drops by 12–13%. On `gte-modernbert-base`, the drop is much larger (40% nDCG@10; 44% Acc@1). This supports the predicted tension: under a single pooled embedding with cosine scoring, allocating margin to reject lexically overlapping meaning-changes competes with broad topical grouping.

**Does the retrieval drop buy identity sensitivity in pooled space?** To measure what compositional sensitivity is obtained *within the pooled space*, we plot cosine-similarity distributions between an original sentence $s$ and a minimally perturbed near-miss $\tilde{s}$ (negation, binding/order, spatial flips). Lower cosine is better: all perturbations are non-identical by construction. Fig. 1 overlays these distributions with 10k held-out NQ positives and negatives.

Two patterns stand out. First, NQ-only fine-tuning (Model A) leaves identity-breaking edits highly similar to the anchor: negation and binding remain near the positive regime, and spatial flips are nearly saturated. Second, introducing structural negatives (Model B) produces *non-uniform* improvements: while it significantly reduces similarity for negation and spatial flips, the gains for binding are less definitive. Despite a lower mean, binding lacks a distinct cluster to separate it from other categories. Thus, while structure-targeted negatives improve sensitivity for specific perturbation classes, they fail to establish a consistent identity margin in pooled cosine space, underscoring the continued necessity of token-interaction verification.

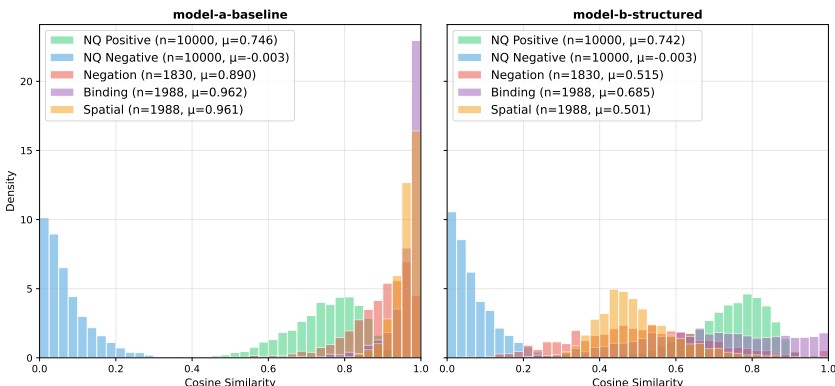

Figure 1: Cosine-similarity distributions between an anchor sentence and a minimally edited near-miss under pooled embeddings. We compare Model A vs. Model B for three perturbation families (negation, binding/order, spatial) and overlay NQ positives/negatives for reference (10k pairs each). Lower is better for near-miss distributions.

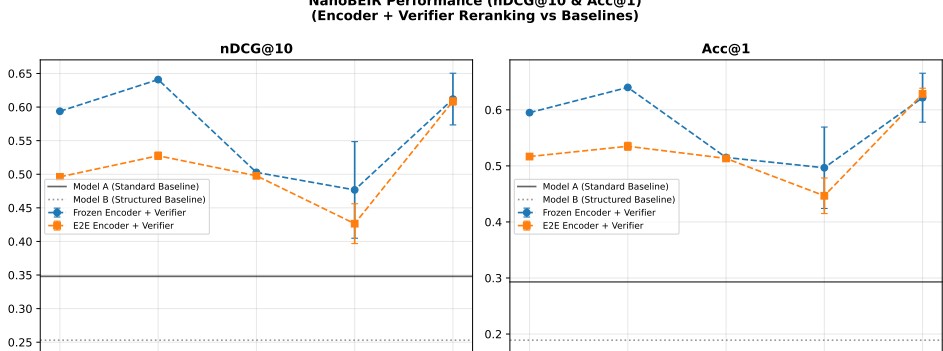

Figure 2: NanoBEIR performance after reranking top-$K$ candidates with $F_k$ under a frozen-encoder (blue) or end-to-end (orange) regime; horizontal lines show encoder-only baselines (Model A and Model B). MaxSim ($F_1$) is the strongest frozen reranker; end-to-end $F_4$ is most competitive.

**Takeaway:** structural negatives partially lower cosine for some edits but reliably hurt out-of-domain retrieval.

### 3.2 HOW SMALL CAN THE VERIFIER BE?

We evaluate the verifier family $\{F_k\}$ from §C.2 operating over token–token cosine maps $M(q, c)$. We compare: (i) **Frozen encoder**, where we train only the verifier, and (ii) **End-to-end**, where we train encoder and verifier jointly. All methods share the same stage-1 candidate generation via pooled cosine; only the stage-2 verifier differs.

**Evaluation 1: reranking on NanoBEIR.** Fig. 2 reports NanoBEIR metrics after reranking the top-$K$ candidates with each verifier. In the **frozen** regime, late interaction $F_1$ (MaxSim) is the strongest and most consistent reranker across metrics; $F_0$ and $F_4$ are often close, while soft alignment $F_2$ is consistently weaker. In the **end-to-end** regime, verifier choice matters more: jointly training with the map-Transformer $F_4$ yields the largest and most reliable gains.

**Evaluation 2: synthetic structural near-miss test.** We evaluate on the held-out 5,964-pair split from §D.1, grouped into **Negation**, **Binding/Order**, and **Spatial**. Fig. 3 plots the mean score assigned

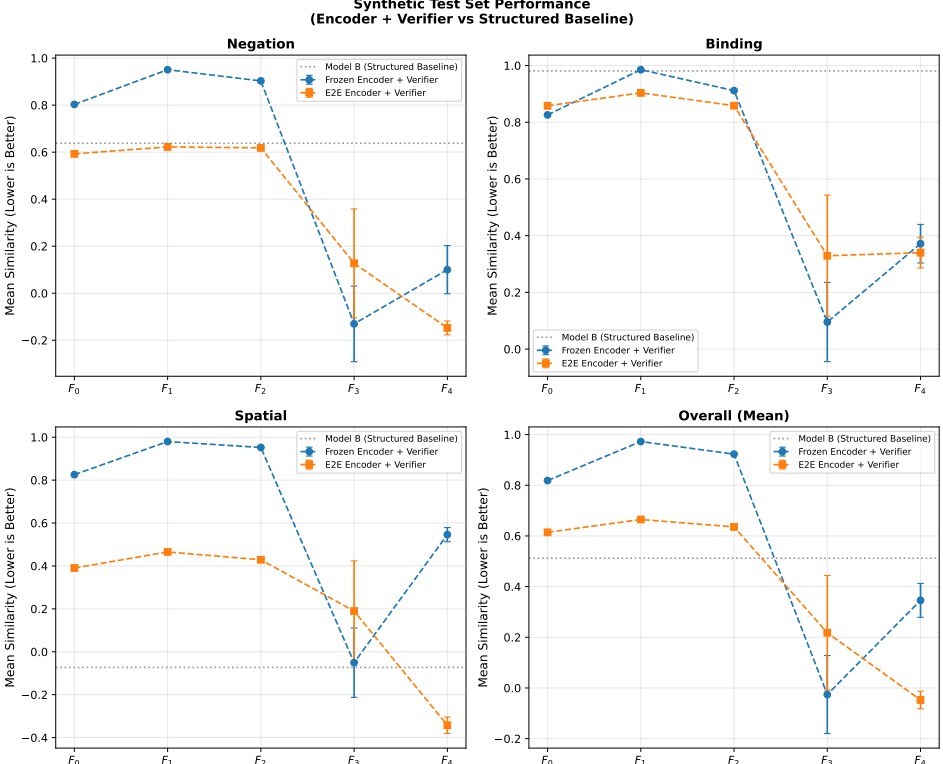

Figure 3: Synthetic structural near-miss test. Mean scores on hard negatives (near-misses); lower is better. The dotted line is pooled cosine from Model B. Simple reductions of $M$ ($F_0$–$F_2$) and MaxSim ($F_1$) score near-misses as highly similar, while topology-aware verifiers ($F_3$, $F_4$) substantially reduce near-miss scores; end-to-end $F_4$ is strongest on spatial flips.

to near-miss pairs (lower is better). The dotted horizontal line shows the pooled-cosine score from the structured encoder baseline (Model B).

**Results.** Comparing Fig. 2 and Fig. 3 highlights a key mismatch. MaxSim ($F_1$) improves benchmark reranking on NanoBEIR but fails to reject structural near-misses, assigning them near-identity scores. Conversely, learned map-based verifiers ($F_3/F_4$) substantially improve near-miss separation, with $F_4$ strongest under end-to-end training, but are not always the top frozen rerankers. This reinforces that if a deployment requires identity-level correctness, verification must be treated as a distinct objective with appropriate data and calibration, rather than assumed to follow from relevance benchmarks.

**Takeaway:** MaxSim is a strong relevance reranker, but identity rejection needs learned map structure.

## 4 DISCUSSION AND CONCLUSION

Pooled-cosine embeddings are a strong *recall* interface for content grouping, but our results support a structural limitation for identity-sensitive matching: injecting identity-focused negatives into a single-vector objective can trade off against out-of-domain relevance retrieval. Token-interaction verification is a principled escape hatch, but relevance reranking (NanoBEIR) and identity rejection are not automatically aligned: MaxSim helps the former while failing the latter, whereas small learned verifiers over similarity maps better enforce compositional identity. This motivates treating identity-sensitive verification as a distinct objective with dedicated data and calibration.

## REPRODUCIBILITY STATEMENT

Complete experimental settings (model architectures, hyperparameters, preprocessing, random seeds, hardware/software versions, and evaluation protocol) are provided in Appendix D. The shared anonymized repository includes the code used to train and evaluate all models, scripts for dataset construction, and the exact dataset splits used in our experiments.

## ETHICS STATEMENT

We adhere to the ICLR Code of Ethics. Our experiments use only publicly available benchmark datasets and automatically constructed structural near-miss examples; we collect no new user data and involve no human subjects. We comply with dataset licenses and will release only license-compliant artifacts. Potential risks include biased retrieval/verification behavior inherited from pretrained models or dataset distributions; we recommend auditing before deployment in sensitive applications.

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

## A   EXPANDED RELATED WORK

**Pooled embeddings and compositional failures.** Single-vector cosine embeddings enable fast ANN retrieval but often under-encode binding, order, and scoped negation; stress tests find strong retrieval despite compositional ablations, suggesting shortcut solutions (Yuksekgonul et al., 2022; Kamath et al., 2023; Hsieh et al., 2023; Alhamoud et al., 2025).

**Geometric analyses and token-interaction remedies.** Kang et al. (2025) show that cosine spaces satisfying basic categorization induce linear superposition, collapsing attribute binding and conflicting with spatial relations and negation; they propose Dense Cosine Similarity Maps and lightweight CNNs over interactions.

**Two-stage retrieval and verification.** Candidate generation plus reranking is standard: cross-encoders compute full interactions, while late interaction retains token structure with the efficient MaxSim operator (Nogueira & Cho, 2019; Nogueira et al., 2020; Khattab & Zaharia, 2020; Santhanam et al., 2022). Sparse expansions (e.g., SPLADE) offer an alternative first-stage representation (Formal et al., 2021).

**Indexing and compression.** ANN systems and quantization are standard for dense retrieval (Johnson et al., 2018; Malkov & Yashunin, 2018; Jégou et al., 2011; Ge et al.).

**Embedding geometry.** Work on anisotropy and cosine similarity supports structured scoring beyond pooled cosine (Ethayarajh, 2019; Steck et al., 2024).

## B  THEORY DETAILS: POOLED-COSINE BRITTLENESS

Many semantic search deployments are *content-relevance* oriented regardless of fine-grained semantic differences. However, several important applications require *identity-sensitive* matching: the system must accept a candidate only if it expresses the same proposition up to paraphrase, rejecting candidates with nearly identical wording but different meaning or intent (see examples in §1). We treat as *non-identical* (near-miss negatives) edits that change: (i) *attribute–head binding* (which modifier applies to which head), (ii) *relations and argument roles/order* (subject/object swaps, attachment changes), or (iii) *negation and scope* (polarity flips or changes in what an operator negates).

### B.1  SINGLE-VECTOR COSINE RETRIEVAL

Let $\mathcal{V}$ be a vocabulary and $\mathcal{S} \subseteq \mathcal{V}^*$ the set of well-formed sentences (or clauses). We study *text-only* embedding-based semantic search systems that map each $s \in \mathcal{S}$ to a single vector and use ANN search to retrieve candidates. We write $q \equiv c$ when $q$ and $c$ express the same proposition.

Let $e_\theta : \mathcal{S} \to \mathbb{S}^{d-1}$ map each sentence to a *unit* vector in $\mathbb{R}^d$.[2] A standard match surrogate is cosine thresholding,

$$\mathsf{accept}_\tau(q, c) \;=\; \mathbf{1}\big[\cos\big(e_\theta(q), e_\theta(c)\big) \geq \tau\big]. \tag{1}$$

This interface enables compact indexes and efficient ANN search, but it enforces a severe bottleneck: all semantics must be encoded into a single direction on the sphere, and the decision depends on a single inner product.

### B.2  WHY POOLED COSINE IS BRITTLE FOR COMPOSITIONAL IDENTITY

Our analysis follows the *ideal-geometry* framework of Kang et al. (2025). They formalize conditions for an "ideal" CLIP-like unit-sphere cosine space and prove these conditions are mutually incompatible: satisfying basic concept categorization forces a linear superposition geometry that cannot also satisfy binding, spatial relations, and negation. We adapt the implication to text-only retrieval; full formal definitions and proofs are in Kang et al. (2025) (and its supplement), and we focus primarily on empirical consequences for text retrieval.

**Content grouping and superposition.** Dense retrievers are typically trained/evaluated so that texts sharing salient content words or topics are closer than texts with disjoint content. Under unit-norm embeddings with cosine scoring, Kang et al. (2025) show that the cosine-optimal representation for a composition that must remain close to its constituents is (approximately) a normalized linear superposition. In text terms, if a sentence expresses salient units $x_1, x_2 \in \mathcal{V}$ and must remain close to each while repelling unrelated content, then

$$e_\theta(x_1 \, x_2) \;\approx\; \frac{e_\theta(x_1) + e_\theta(x_2)}{\|e_\theta(x_1) + e_\theta(x_2)\|}. \tag{2}$$

Superposition is commutative; without additional structure at scoring time, it naturally encourages invariances that erase binding and role information.

**Minimal identity constraints.** For identity-sensitive matching, we would like paraphrases $q^+ \equiv q$ to be closer than minimally edited near-misses $q^- \not\equiv q$ by a margin:

$$\cos(e_\theta(q), e_\theta(q^+)) \;\geq\; \cos(e_\theta(q), e_\theta(q^-)) + \gamma. \tag{3}$$

Near-misses include (i) binding swaps, (ii) role/order reversals, and (iii) negation/scope flips.

**Assumptions.** We isolate the interface shared by most embedding retrievers:

**A1** *Single pooled key:* each sentence is represented by one unit vector in $\mathbb{S}^{d-1}$.

**A2** *Cosine scoring:* decisions depend only on cosine similarity between pooled keys.

**A3** *No token interactions at score time:* the scorer has no access to token–token alignments beyond what is compressed into the pooled key.

---

[2]We write $\mathbb{S}^{d-1} = \{u \in \mathbb{R}^d : \|u\|_2 = 1\}$.

**Theorem 1** (Informal pooled-cosine brittleness for compositional identity). *Under A1–A3, any encoder family that enforces nontrivial content grouping (compositions remain close to their constituents with margin) necessarily admits clause pairs that differ only by (i) attribute binding, (ii) relational roles/order, or (iii) negation/scope, yet cannot be simultaneously separated from identity-preserving paraphrases by a fixed cosine margin.*

*Justification* Content grouping implies an approximately additive/superpositional placement (Lemma 1 in Kang et al. (2025)); commutativity yields binding collapse (Lemma 2) and analogous invariances for role/order. When one additionally enforces natural cosine behavior for negation, Kang et al. (2025) derive further contradictions. We omit the full formalization for text and refer to Kang et al. (2025) for complete proofs.

**Corollary 1** (Threshold brittleness). *If A1–A3 hold and content grouping has margin $\gamma_{\text{cont}} > 0$, then for any fixed threshold $\tau$ there exist minimally edited near-miss pairs $(q, c)$ (binding swap, role reversal, or scoped negation flip) such that Eq. equation 1 incurs either a false accept or a false reject at a scale comparable to $\gamma_{\text{cont}}$.*

A practical implication is a *retrieval–composition tension*: if we insist on a single pooled key and cosine as the only scoring mechanism, encoding fine-grained structure competes with the angular budget used for coarse content grouping. In §3, we test whether structure-targeted hard negatives produce this trade-off in text-only dual-encoder training.

## C  VERIFIER DEFINITIONS AND ARCHITECTURES

Theorem 1 points to an interface mismatch: the bottleneck is not necessarily the token representations themselves, but the fact that the final decision collapses everything into one cosine score. A natural remedy—already prevalent in IR—is a two-stage pipeline: use pooled embeddings for high-recall candidate generation, then *verify* (or rerank) with token-level interactions (Nogueira & Cho, 2019; Khattab & Zaharia, 2020).

### C.1  TWO-STAGE RETRIEVAL WITH TOKEN-LEVEL VERIFICATION

**Stage 1 (candidate generation).** A transformer encoder produces contextual token embeddings $H_\theta(s) = [h_1, \ldots, h_{m(s)}] \in \mathbb{R}^{m(s) \times d}$. We pool to a unit key $e_\theta(s) \in \mathbb{S}^{d-1}$ (CLS/mean/EOS) and retrieve top-$K$ candidates with ANN under cosine similarity.

**Stage 2 (verification).** For a query $q$ and candidate $c$ with token embeddings $Q = [q_1, \ldots, q_m]$ and $C = [c_1, \ldots, c_n]$, define the token similarity map

$$M(q, c) \in [-1, 1]^{m \times n}, \qquad M_{ij}(q, c) = \cos(q_i, c_j). \tag{4}$$

Here $\phi$ denotes elementwise normalization/clipping of $M$, and $\psi$ patches (or flattens) the map into a sequence for the Transformer. A verifier consumes $M(q, c)$ (optionally with positional information) and outputs a scalar score $F(q, c)$ used for gating or reranking.

### C.2  A SPECTRUM OF LIGHTWEIGHT VERIFIERS

We study verifiers $\{F_k\}$ that vary in expressivity/cost while remaining far cheaper than full cross-encoding over long corpora. All verifiers operate on $M$ after stage-1 retrieval.

$$F_0(q,c) = \frac{1}{mn} \sum_{i=1}^{m} \sum_{j=1}^{n} M_{ij} \qquad \text{(global average)} \qquad (5)$$

$$F_1(q,c) = \frac{1}{m} \sum_{i=1}^{m} \max_j M_{ij} \qquad \text{(late interaction / MaxSim)} \qquad (6)$$

$$F_2(q,c) = \frac{1}{m} \sum_{i=1}^{m} \sum_{j=1}^{n} A_{ij}(q,c)\, M_{ij} \qquad \text{(soft alignment with positional bias)} \qquad (7)$$

$$F_3(q,c) = \text{MLP}\Big(\text{CNN}_{k \times k}\big(\phi(M)\big)\Big) \qquad \text{(tiny CNN over } M\text{)} \qquad (8)$$

$$F_4(q,c) = \text{MLP}\Big(\text{Transformer}\big(\psi(\phi(M))\big)_{[\text{CLS}]}\Big) \qquad \text{(tiny Transformer over patches of } M\text{)} \qquad (9)$$

where $A(q,c)$ is a row-stochastic alignment matrix:

$$A_{ij}(q,c) = \frac{\exp\big((M_{ij}(q,c) - \lambda|i-j|)/\tau\big)}{\sum_{k=1}^{n} \exp\big((M_{ik}(q,c) - \lambda|i-k|)/\tau\big)}. \qquad (10)$$

### C.3 WHY TOKEN INTERACTIONS HELP

The pooled-cosine bottleneck collapses many compositions because it discards token topology. By contrast, $M(q,c)$ preserves which tokens align and *where* those alignments occur. Verifiers that only aggregate $M$ with permutation-symmetric statistics (e.g., $F_0$, and to a large extent $F_1$) can still behave like bag-of-words matchers and remain insensitive to binding or role swaps. Injecting positional structure (as in $F_2$) and learning local/global patterns over $M$ (as in $F_3/F_4$) breaks these symmetries, allowing the verifier to detect order-preserving diagonals, swapped alignments, and systematic mismatches induced by negation cues. This mirrors the core insight of DCSMs in Kang et al. (2025), specialized here to text–text matching.

## D EXPERIMENTAL DETAILS

This section summarizes the datasets, model variants, training setup, and evaluation protocol needed to reproduce our results.

### D.1 DATA

**Baseline training data (Natural Questions).**  We fine-tune dual encoders on 100,000 triplets sampled from Natural Questions (Kwiatkowski et al., 2019) using the standard (`anchor`, `positive`, `negative`) format.

**Structural hard negatives.**  We augment training with *structural near-misses*: lexically high-overlap pairs whose meaning differs due to (i) negation/scope flips, (ii) binding/order changes, or (iii) spatial relation flips. We construct 9,940 pairs per category (29,820 total) and convert each pair $(s_1, s_2)$ into a triplet $(s_1, s_1, s_2)$ so the model must repel the near-miss while keeping the anchor fixed. We split pairs 80/20 and use the held-out split (5,964 pairs) for synthetic evaluations.

The final structured-training mixture contains 123,856 triplets, where 23,857 (19.2%) are structural-negative triplets and the remainder are standard NQ triplets. We drop null/placeholder rows, filter sentences shorter than 20 characters, and truncate/pad to 128 tokens.

### D.2 MODELS

**Stage-1 candidate generators (dual encoders).**  We evaluate four backbones: `sentence-transformers/all-MiniLM-L6-v2`, `sentence-transformers/all-MiniLM-L12-v2`, `thenlper/gte-small`, and

`Alibaba-NLP/gte-modernbert-base`. We use the default pooling method of each encoder, max length 128, and unit-normalized pooled embeddings with cosine scoring. MiniLM and `gte-small` use 384-d pooled embeddings; other backbones use their native embedding dimensions.

**Stage-2 verifiers.** Verifiers consume token–token cosine maps $M(q, c)$ and output a scalar score for reranking/gating (Appendix C). We evaluate $F_0$–$F_4$ as defined in Appendix C. Learned verifiers use small networks over $M$ (a tiny CNN for $F_3$ and a tiny Transformer for $F_4$).

### D.3 TRAINING

**Encoder training objective.** We fine-tune using SentenceTransformers' MultipleNegativesRank-ingLoss with temperature $\tau{=}0.1$, optimized with AdamW and a linear warmup/decay schedule.

**Key hyperparameters.** Unless otherwise stated: learning rate $2{\times}10^{-5}$ (scaled by model size in code), weight decay 0.01, batch size 64 (and 128 in selected runs), warmup ratio 0.1, gradient accumulation 1, fp16/bf16 precision. We fix wall-clock training time per backbone and set steps based on measured throughput.

**Verifier training.** We compare (i) **Frozen** (train verifier only) and (ii) **End-to-end** (train encoder+verifier jointly). Verifier LR is $1{\times}10^{-4}$; end-to-end encoder LR is $1{\times}10^{-5}$ (scaled by model size in code). Batch size is 128 for $F_0$–$F_2$ and 32 for $F_3$–$F_4$. We early-stop with patience 5000 steps on nDCG@10.

**Random seeds.** Primary seed is 42. Multi-seed results use seeds $\{42, 43, 44\}$.

### D.4 EVALUATION PROTOCOL

**Retrieval benchmarks.** We evaluate zero-shot retrieval on NanoBEIR (`lightonai/NanoBEIR-en`) and report mean performance across datasets. We report nDCG@10 and Acc@1 in the main paper (additional metrics are computed in code).

**Two-stage evaluation.** Stage 1 retrieves top-$K{=}100$ candidates using pooled-cosine ANN. Stage 2 (optional) reranks/gates the top-$K$ using a verifier score. Evaluation batch size is 32.

### D.5 COMPUTE AND SOFTWARE

We run on GPUs with $\geq$24GB VRAM (tested on NVIDIA L4 and A10-class hardware). Typical training time is $\sim$4 minutes per configuration; the full experiment suite runs in $\sim$2–3 hours. We use Python 3.10 with PyTorch, HuggingFace Transformers, SentenceTransformers, and BEIR; exact versions are pinned in the released environment files.

