# OpenReview forum: "Training for Compositional Sensitivity Reduces Dense Retrieval Generalization"
_ICLR.cc/2026/Workshop/Sci4DL — Sci4DL 2026_

### Official Review · Reviewer_snDU · 2026-02-15

**Fit:** 2
**Significance:** 2
**Confidence:** 2

**Summary:**

This work addresses the problem of identity-level matching in retrieval using cosine distance over pooled embeddings. Specifically, the authors focus on text retrieval and demonstrate that small structural modifications (termed structural near-misses), which dramatically change the meaning of a text, do not significantly affect cosine similarity — much like paraphrases, which preserve meaning. Furthermore, the authors show that fine-tuning with negative samples designed to force embeddings to capture these distinctions improves performance on such cases but degrades retrieval generalization. They refer to this phenomenon as *retrieval–composition tension*. Additionally, they demonstrate that incorporating trainable fine-grained verifiers further improves retrieval on structural near-miss tests.

**Strengths:**

1. The results are sound and empirically reveal an important characteristic of pooled dense embeddings when used with cosine distance.
2. The work adheres to reproducibility standards and provides the code for all conducted experiments.

**Suggestions:**

1. The paper is not self-contained, at least in the main body. It should provide brief definitions for key concepts, e.g., MaxSim. Moreover, the study lacks a clearly stated motivation or research question.
2. Figure 2: The line plot represents results for different re-rankers $F_k$​. Since these re-rankers are distinct models, a bar plot (or similar) would be more appropriate, as the current line plot misleadingly implies a continuous relationship between them. Additionally, the subscript $k$ in $F_k$​ suggests link to top−K candidate value, like the verifier was bound to the fixed number of retrieved items $k$. I recommend using full names for the re-rankers instead.
3. The language of the paper is at times vague, with overly condensed sentences and unexplained abbreviations, which hinders comprehension. If space is limited, I encourage the authors to reduce plot sizes or present only $nDCG@10$ in the main text, moving $Acc@1$ to the appendix.

---

### Official Review · Reviewer_83BH · 2026-02-27

**Fit:** 3
**Significance:** 2
**Confidence:** 2

**Summary:**

The submission focuses on dense retrieval, delving into the retrieval-composition tension. Starting from the observation that dense retrieval struggles to encode compositional distinctions, the authors 1) investigate training with structure-targeted hard negatives, showing that it improves the compositional sensitivity of the model but degrades out-of-domain retrieval; 2) add a variety of verifiers to the pipeline to reject structural near-misses, showing that a *learned* verifier is needed to enforce compositional identity.

**Strengths:**

The submission is clearly structured, tackles an interesting question, and presents extensive experiments to investigate the retrieval-composition tension in dense retrieval.

**Suggestions:**

I found the paper quite difficult to read for non-specialists in retrieval:
- Could the authors provide examples of retrieval tasks where compositional sensitivity is important, for example in the Introduction?
- Defining the words "structural near-miss", "compositional sensitivity", "structure-targeted hard negative" in the main body of the paper, and expanding "ANN" (line 64) and "NQ" (line 82) at least once, would be helpful for the non-specialist readers
- What is the formula for the pooling in the experiments? (I would suggest to at least refer to the Appendix where it is explained)

Other comments and questions:
- Adding a literature review about compositional sensitivity in dense retrieval and methods to improve it, if this has been investigated before, would be crucial to help readers identify the open questions the submission is answering.
- In Figures 2 and 3, I think the names of $F_0, ..., F_4$ (not the formulas) should appear in the main text, and not only in the Appendix, to increase the readability of the figures.
- I would suggest merging Figures 2 and 3, keeping only one of the plots of Fig. 2, and only the mean plot of Fig. 3, so that the curves can be compared more easily. This would make the retrieval-composition tension more visible. The other plots could be moved in the Appendix, as their conclusions are similar.
- It appears from Figures 2 and 3 that F4 is the best end-to-end choice for both retrieval and compositional sensitivity. If the authors agree with this observation, I think it deserves to be included in the Takeaway (which currently focuses only on the frozen setting).
- I am curious: is there an intuition as to why the frozen pipeline performs better than the end-to-end one on NanoBEIR? This seems rather counterintuitive to me.
- In Appendix C, lines 416-417, the authors use $Q$ and $C$ while equation (4) uses $q$ and $c$ (as well as the formula line 66).

---

### Official Review · Reviewer_mtCi · 2026-02-27

**Fit:** 3
**Significance:** 2
**Confidence:** 3

**Summary:**

This paper investigates the fundamental tension between compositional sensitivity (distinguishing structural differences like negation, role swaps, and attribute binding) and topical generalization (broad relevance matching) in single-vector dense retrieval models. Motivated by geometric impossibility results regarding unit-sphere embeddings (Kang et al., 2025), the authors empirically demonstrate that training dual-encoders with structure-targeted hard negatives creates a "zero-sum game": while structural sensitivity improves, out-of-domain retrieval performance on NanoBEIR degrades significantly (dropping 8–9% for small backbones and up to ~40% for medium ones). To resolve this "resolution loss," the authors propose a two-stage pipeline: using pooled cosine similarity for efficient recall, followed by a learned verifier that analyzes token–token similarity maps. The study highlights that while MaxSim (late interaction) excels at relevance reranking, it fails to reject structural near-misses. In contrast, a lightweight Transformer verifier trained to recognize topological patterns in similarity maps (e.g., diagonal consistency) successfully achieves identity-level matching without sacrificing retrieval efficiency.

**Strengths:**

The submission offers a rigorous and insightful examination of the limitations of dense retrieval, making several significant contributions:

- **Identification of a Fundamental Trade-off**: The paper provides compelling empirical evidence for a "zero-sum game" between compositional sensitivity and topical generalization. By demonstrating that training for structural precision (e.g., distinguishing role swaps) causes significant performance drops on out-of-domain benchmarks (up to ~40% nDCG@10 drop for medium backbones), the authors challenge the prevailing assumption that hard-negative mining is universally beneficial.
- **Strong Theoretical Grounding**: The work effectively adapts recent geometric theories (Kang et al., 2025) to the text-only domain. It offers a sound theoretical explanation for why single-vector models fail at binding and negation: the "resolution loss" inherent in compressing variable-length reasoning into a fixed-size vector forces a commutative geometry that erases order information.
- **Critical Evaluation of Late Interaction (MaxSim)**: The paper makes a valuable distinction between "relevance ranking" and "identity verification." It explicitly demonstrates that widely used late-interaction methods like MaxSim (F1), while effective for relevance, fail to reject structural near-misses because they remain largely permutation-symmetric.
- **Novel and Effective Methodology**: The proposal to treat token-token similarity maps as images and use lightweight "vision-style" verifiers (Tiny CNN/Transformer) is both innovative and practically demonstrated. The results show that these learned verifiers (F3, F4) successfully detect topological patterns (like diagonals or swapped alignments) to identify binding and negation errors where traditional methods fail.

**Suggestions:**

To further improve the paper, I recommend the following:
1. Investigate the Disproportionate Collapse of ModernBERT: The performance drop for GTE-ModernBERT-base is extreme (approx. -40% nDCG@10) compared to the smaller models (approx. -8%). While the authors attribute this to the "zero-sum" geometric tension where capacity is monopolized by structural differentiation, such a drastic divergence invites questions about hyperparameter sensitivity or catastrophic overfitting specific to the medium backbone. An ablation study checking if this drop can be mitigated (e.g., by adjusting the mixing ratio of structural negatives or the learning rate for the medium model) would clarify whether this is a fundamental geometric inevitability or a training instability.
2. Address Context Length and Scalability: The current experiments truncate sequences to 128 tokens. Since the proposed token similarity map (M) and the subsequent verifier attention scale quadratically (N×M), the computational cost for standard passage retrieval lengths (e.g., 512 tokens) could be significant. It would be valuable to discuss how the "Tiny Transformer" verifier (F4) handles longer contexts or if patching/pooling strategies are robust enough for longer documents without prohibitive memory costs.
3. Provide Quantitative Latency Metrics: The paper positions the verifiers (F3,F4) as "lightweight" and "far cheaper than full cross-encoding". However, specific latency (ms/query) or throughput numbers are missing. A table comparing the inference time of the proposed two-stage pipeline against a standard Cross-Encoder reranker (and MaxSim) would substantiate the efficiency claims and help practitioners evaluate the trade-off between the high structural accuracy of F4 and its cost.
4. Comparison with Cross-Encoders: While the paper compares the verifiers against MaxSim (Late Interaction), a comparison against a standard Cross-Encoder (e.g., a BERT-based reranker) is missing. Even if the Cross-Encoder is slower, establishing it as an "upper bound" for structural sensitivity would help contextualize how close the proposed map-based verifiers (F4) come to achieving full-attention performance using only the frozen similarity map.

---

### Meta-Review · Area_Chair_nkKy · 2026-02-28

**Recommendation:** Accept

**Metareview:**

Reviews were positive, and I recommend acceptance.

---

### Decision · Program_Chairs · 2026-03-02

Accept